# Diversity and inclusion for the *All of Us* research program: A scoping review

**Brandy M. Mapes** [1]*, **Christopher S. Foster**[2], **Sheila V. Kusnoor**[3], **Marcia I. Epelbaum**[3], **Mona AuYoung**[4], **Gwynne Jenkins**[2], **Maria Lopez-Class**[2], **Dara Richardson-Heron**[2], **Ahmed Elmi**[2], **Karl Surkan**[5], **Robert M. Cronin** [6], **Consuelo H. Wilkins**[7], **Eliseo J. Pérez-Stable**[8], **Eric Dishman**[2], **Joshua C. Denny**[9], **Joni L. Rutter** [10]*, **the *All of Us* Research Program**¶

**1** Vanderbilt Institute for Clinical and Translational Research, Vanderbilt University Medical Center, Nashville, Tennessee, United States of America, **2** Office of the Director, National Institutes of Health, Bethesda, Maryland, United States of America, **3** Center for Knowledge Management, Vanderbilt University Medical Center, Nashville, TN, United States of American, **4** Scripps Whittier Diabetes Institute, Scripps Health, San Diego, California, United States of American, **5** Massachusetts Institute of Technology, Cambridge, Massachusetts, United States of America and *All of Us* Research Program Participant Representative, **6** Department of Biomedical Informatics, Medicine, and Pediatrics, Vanderbilt University Medical Center, Nashville, Tennessee, United States of America, **7** Department of Medicine, Vanderbilt University Medical Center, Nashville, Tennessee, United States of America, **8** National Institute on Minority Health and Health Disparities, Bethesda, Maryland, United States of America, **9** Departments of Biomedical Informatics and Medicine, Vanderbilt University Medical Center, Nashville, Tennessee, United States of America, **10** National Center for Advancing Translational Sciences, National Institutes of Health, Bethesda, Maryland, United States of America

¶ Membership of the the All of Us Research Program is provided in the Acknowledgments.
* brandy.mapes@vumc.org (BMM); joni.rutter@nih.gov (JLR)

**Data Availability Statement:** All relevant data are within the paper and its Supporting Information files.

## Abstract

The *All of Us* Research Program (*All of Us*) is a national effort to accelerate health research by exploring the relationship between lifestyle, environment, and genetics. It is set to become one of the largest research efforts in U.S. history, aiming to build a national resource of data from at least one million participants. *All of Us* aims to address the need for more diversity in research and set the stage for that diversity to be leveraged in precision medicine research to come. This paper describes how the program assessed demographic characteristics of participants who have enrolled in other U.S. biomedical research cohorts to better understand which groups are traditionally represented or underrepresented in biomedical research. We 1) reviewed the enrollment characteristics of national cohort studies like *All of Us*, and 2) surveyed the literature, focusing on key diversity categories essential to the program's enrollment aims. Based on these efforts, *All of Us* emphasizes enrollment of racial and ethnic minorities, and has formally designated the following additional groups as historically underrepresented: individuals—with inadequate access to medical care; under the age of 18 or over 65; with an annual household income at or below 200% of the federal poverty level; who have a cognitive or physical disability; have less than a high school education or equivalent; are intersex; identify as a sexual or gender minority; or live in rural or non-metropolitan areas. Research accounting for wider demographic variability is critical. Only by ensuring diversity and by addressing the very barriers that limit it, can we position *All of Us* to better understand and tackle health disparities.

**Funding:** Authors were supported by the following grants through the National Institutes of Health, Office of the Director (https://allofus.nih.gov/): U2COD023196 (Mapes, Cronin, Wilkins, Denny), OT2OD023132 (Mapes, Kusnoor, Epelbaum, Cronin, Wilkins, Denny), and U24 OD023176 (AuYoung). Additional support was received through the National Center for Advancing Translational Sciences (https://ncats.nih.gov/): UL1TR002550 (AuYoung).

**Competing interests:** The authors have declared that no competing interests exist.

# Introduction

The *All of Us* Research Program (*All of Us*), funded by the National Institutes of Health (NIH), is a national effort to accelerate health research by exploring the relationship between lifestyle, environment, and genetics [1]. The program is set to become one of the largest health care research efforts in U.S. history, aiming to build a longitudinal resource of multiple data types and biosamples from at least one million participants.

A main goal of the NIH Precision Medicine Initiative (PMI) Working Group [2] was that *All of Us* "should broadly reflect the diversity of the U.S." As stated in the 21<sup>st</sup> Century Cures Act, *All of Us* will "ensure inclusion of a broad range of participants, including consideration of biological, social, and other determinants of health that contribute to health disparities." [3] The recognition that not all groups have benefited equally from research is a major step towards health equity. A key contribution that *All of Us* can make is setting the stage for more diverse participation in precision medicine research to come. We describe the efforts made to date to guarantee that *All of Us* reflects the broad diversity of the U.S. and enables precision medicine research for all.

To ensure that *All of Us* is rigorous in its efforts to enroll a diverse group of one million participants, we assessed the demographic characteristics of other U.S. biomedical research cohorts. To better understand which groups are represented or underrepresented in biomedical research, we 1) conducted a broad survey of the literature focused on ten demographic categories essential to the program's diversity aims, and 2) reviewed the enrollment characteristics of national cohort studies similar to *All of Us*. By identifying these populations, we can understand the barriers and gaps in enrollment of key underrepresented groups. Based on previous evaluations of diversity and inclusion by race and ethnicity, non-White groups have already been established as underrepresented. [4–6] It is important to note here a distinction between minority health and underrepresentation of other groups, especially with regards to health disparities research and biomedical research representation. Minority health refers to "health characteristics and attributes of racial and/or ethnic minority groups (defined by Office of Management and Budget), who are socially disadvantaged due in part by being subject to potential discriminatory acts." [7] Historically, women have been underrepresented as well, though their proportion now exceeds 50% in NIH-funded research. [8] In addition to racial and ethnic diversity in biomedical research, this work proposes that researchers examine other demographic and social determinants when considering inclusion of diverse participant populations. This review established a landscape of demographic characteristics and social determinants meant to guide *All of Us* towards measuring diversity among its participants.

## A Guide for diversity inclusion

### Defining underrepresentation for *All of Us* and stakeholder input

*All of Us* has relied on designated definitions of diversity guided by the leading authorities on health disparities. In addition, we have considered the inclusion of groups that may be underrepresented in biomedical research, specifically, through consultation with a variety of resources and stakeholders, as follows. The National Institute on Minority Health and Health Disparities (NIMHD), congressionally mandated to designate disparity groups, [9] has designated the following groups as most affected by health disparities in the U.S.: Blacks/African Americans, Hispanics/Latinos, American Indians/Alaska Natives, Asian Americans, Native Hawaiians and other Pacific Islanders, socioeconomically disadvantaged populations, underserved rural populations, and sexual and gender minorities. We also used the guidelines for including women and minorities in clinical research set forth by the NIH [10], as well as a

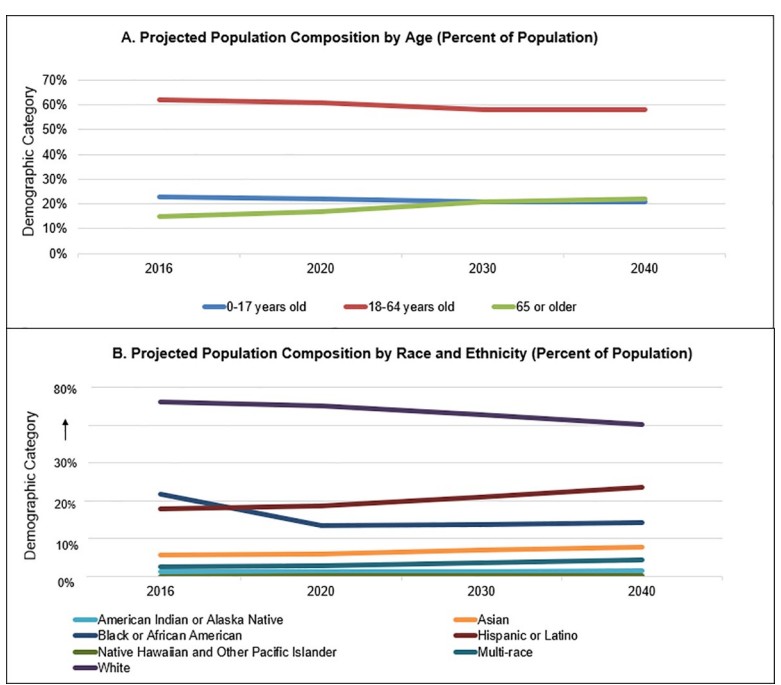

**Fig 1.** (A-B). Projected Age, Race, & Ethnicity, and Sex Composition of the United States through 2040.

review of the U.S. Census data (Fig 1) [11]. We sought input from NIH leadership and associated stakeholders, including the NIMHD; the Office of Research on Women's Health; the Sexual and Gender Minority Research Office; the National Human Genome Research Institute; the National Institute of Child Health and Human Development; the National Institute on Aging; the Office of Extramural Research; and the *All of Us* Research Program. Lastly, input was obtained from the *All of Us* Participant Provided Information Committee (PPIC), an *All of Us* governance body who developed the program's baseline demographic survey that uses existing measures from other national surveys. [1] A diverse team of PPIC experts, including *All of Us* participant representatives, and *All of Us* participant engagement teams vetted these measures to ensure appropriate terminology and phrasing and ensured consistency in terminology for guiding the diversity categories developed here, some of which are being implemented as of this writing. We sought to examine if these major groups have historically been underrepresented in the biomedical research literature, including clinical trial studies, and whether these categories would help measure the impact of the All of Us Research Program for these groups.

Based on these major inputs, *All of Us* prioritized nine essential diversity categories to complement racial and ethnic minority inclusion. An important point to note is that, aside from racial/ethnic minorities and sex assigned at birth, the other categories have not been broken out demographically in most other studies. Therefore, it is not explicitly known whether they are truly underrepresented in biomedical research. Explicitly defining and evaluating all these demographic characteristics will be important for understanding their impact on social determinants of health. These include (listed alphabetically): access to care, age, annual household income, disability, educational attainment, gender identity, geography (i.e. rurality), sex assigned at birth, and sexual orientation. We anticipate that eventually 75% of the *All of Us* cohort will be from historically underrepresented groups in biomedical research and expect nearly half of this diversity to be from racial or ethnic minorities. [1] Of the nine categories, six

are included in the NIMHD definitions of "disparity populations," three are related to socio-economic status, two to sexual and gender minorities, and one to rurality. For each category, we verified a designation of groups as either historically represented or underrepresented through a review of the literature.

## Source selection

We established a team with pertinent expertise to plan and execute the review. The team included representatives from the NIH and participating *All of Us* institutions with expertise in library and information science, enrollment monitoring and reporting, underserved populations, health disparities research, and strategic planning for large-scale national consortia.

Searches of English language articles were conducted using PubMed and Google Scholar along with a mixture of Medical Subject Headings (MeSH) terms and keywords relevant to the nine diversity categories (S1 Table for search strategies and updated retrieval counts). Excluded from the results were duplicate articles, those not relevant to human subjects research, and those that did not contain data to either support or refute justification for considering designation of representation. Meta-analyses and systematic review articles published since 2012, specific to biomedical cohort studies conducted in the United States, and emphasizing results or conclusions specific to the categories of interest were prioritized for review by a subgroup of the authors. We also used handsearching and consulted the subject matter experts on our team to identify additional potentially relevant white papers and unpublished reports. Examples of ways evidence about representation in research was assessed included: evaluation of data from existing U.S. biomedical research studies and trials, evidence depicting how study eligibility criteria contributed to underrepresentation of these groups, and evidence outlining barriers to participation unique to these groups. Results were refined to include a small set of distinctive articles that emphasized findings specific to underrepresentation of the key groups (i.e., why they were considered underrepresented, whether the groups were included in study recruitment, data analysis, or conclusions). The literature was reviewed over the course of several months in 2017 through 2018 (S2 Table for final counts of screened and selected articles, S2 File for selected bibliography).

## Complementary assessment of core diversity characteristics for similar cohort studies

Additionally, a selection of other national cohort studies was evaluated for demographic characteristics benchmarking a diversity baseline for *All of Us*. We included cohort studies with an enrollment of >75,000 and a study design or scope like *All of Us*. Those selected were U.S. based except for one international study, the U.K. Biobank. [12] Information was collected using publicly available sources including websites, datasets, and publications. When demographic information was not readily searchable, we made attempts to gather the information from study authors.

Of the national cohort studies evaluated, we identified fifteen as most relevant to informing *All of Us* based on the aims of the research and mixture of data types collected from participants. Five (33%) were still enrolling participants at the time of our review, and seven (47%) had begun enrollment in 2006 or later. We confirmed that thirteen (87%) collected questionnaire data, collected biospecimens, and linked to external data sources. Eight (53%) collected electronic health records. We also prioritized review of diversity categories including age, educational attainment, household income, U.S. geographic region, disability, race and ethnicity, sex assigned at birth, and sexual and gender minority status. See Fig 2 for a brief demographic snapshot of the cohorts we evaluated.

| | BioVU | California Teachers Study | Cancer Prevention Studies Baseline II | Cancer Prevention Study III | Kaiser Permanente Research Program on Genes Environment and Health | Kaiser Permanente Southern California Children's Health Study | The Millennium Cohort Study | Million Veteran Program Active Duty Cohort | Multiethnic Cohort Study of Diet and Cancer | NIH-AARP Diet and Health Study | Nurses' Health Study I | Nurses' Health Study II | Prostate, Lung, Colorectal and Ovarian Cancer Screening Trial | UK BioBank | Women's Health Initiative |
|---|---|---|---|---|---|---|---|---|---|---|---|---|---|---|---|
| Enrollment Dates | ongoing | 1995-1996 | ongoing | 2006-2013 | 2008- ongoing | 2007-2009 | ongoing | ongoing | 1993-1997 | 1995-1996 | 1976 | 1989 | 1993-2001 | 2006-2010 | 1994-2015 |
| Total | 249,208 | 133,479 | 1,185,296 | 303,682 | 270,570 | 920,034 | 77,047 | 397,104 | 215,251 | 545,784 | 121,700 | 116,430 | 154,898 | 503,944 | 93,676 |
| **Demographics Characteristics** | | | | | | | | | | | | | | | |
| **Age** | | | | | | | | | | | | | | | |
| 0-17 | 8% | * | * | * | * | * | * | * | * | 0% | * | * | * | 0% | * |
| 18-64 | 58% | <66% | * | 100.0% | * | * | * | * | * | 63% | 100% | 100% | 64% | 100% | * |
| 65+ | 34% | <17% | * | 0.0% | * | * | * | <32% | * | 37% | * | * | 36% | 0% | * |
| **Education Attainment** | | | | | | | | | | | | | | | |
| < High School/GED | * | * | 15% | 1% | * | * | 6% | * | * | 6% | * | * | * | 30% | 5% |
| > High School/GED | * | * | 85% | 97% | * | * | 94% | * | * | 91% | 100% | 100% | * | 70% | 95% |
| **Race/Ethnicity** | | | | | | | | | | | | | | | |
| White | 80% | 87% | 93% | 83% | * | 21% | 69% | 77% | 23% | 93% | 97% | 92% | 86% | 94% | 83% |
| African American | 10% | 3% | 4% | 5% | * | 8% | 14% | 14% | 16% | 4% | * | * | 5% | 1% | 8% |
| Asian or Pacific Islander | 2% | 3% | >1% | * | * | 7% | 8% | * | 33% | 1% | * | * | * | 1% | 3% |
| Hispanic/Latino | 3% | 4% | >1% | 8% | * | 51% | 6% | * | 22% | 2% | * | * | * | 0% | 4% |
| **Sex** | | | | | | | | | | | | | | | |
| Male | 43% | * | 43% | 23% | * | 50% | 73% | 92% | 30% | 59% | * | * | 49.5% | 45% | * |
| Female | 57% | 100% | 57% | 77% | * | 50% | 27% | 8% | 70% | 41% | 100% | 100% | 50.5% | 54% | 100% |

Data presented were collected September - November 2017; data may not reflect current demographics for those studies still actively enrolling. * indicates instances in which information is unknown, not applicable, not collected, not reported, or not part of the study eligibility criteria. Some totals do not equal 100% because of instances in which demographic data is missing and/or not reported.

**Fig 2. Core demographic characteristics of similar cohorts evaluated to support development of the *All of Us* guide for diversity and inclusion.**

## Results

Roughly 645 articles were screened throughout the literature review and 170 were selected as significant for informing the *All of Us* Guide for Diversity and Inclusion (S2 Table for exact counts by search category). Based on the body of literature and consultation with stakeholders, *All of Us* emphasizes inclusion of racial and ethnic minorities and has prioritized the following additional groups as being possibly underrepresented in biomedical research: individuals with inadequate access to medical care; <18 or >65 years; with an annual household income at or below 200% of the federal poverty level (FPL); who have a cognitive or physical disability; or have less than a high school education or equivalent. Also included are those individuals who: are intersex; identify as a sexual or gender minority; or live in rural or non-metropolitan areas. Including the racial and ethnic groups, Table 1 shows historically represented and underrepresented characteristics of each category and how they are measured by the survey responses. [13] Researchers interested in studying specific subsets of these groups (for example, individuals < 3 or >80), are able to do so based on the way in which these demographic data have been collected.

These designations are supported by assessing the other national cohort studies. Historically underrepresented groups, where we were able to obtain data, were underrepresented across all fifteen studies evaluated. Annual household income was unknown, not collected, or not reported for thirteen (87%). Of those reporting income, one (7%) is U.S. based and only a small portion of the participants had an annual household income under $25,000. [15] Educational attainment data were obtained for seven (47%) cohort studies. Percentage of participants with less than a high school degree or GED ranged from 1% to 15% for those U.S. based studies. [15–18] We confirmed inclusion of older adults in only five (33%) studies and their representation ranged from <17% to 37%. [16,19–22] Participants under 18 were enrolled in two (13%) studies. [22,23] Only one (7%) study reported on representation of participants with a disability; these participants made up 18% of that total sample. [17]

Fourteen (93%) studies reported demographic representation of participants by sex. Males were not eligible for four of the studies. Ten (67%) were eligible to both males and females and reported on sex. Of those ten, females were overrepresented in five (50%). [12,18,21,22,24] None of the fifteen studies reported intersex. Five (33%) reported participants' geographic region. Those mostly recruited from only one U.S. geographic region though two (13%) reported geographic dispersity stretching across at least four regions. [15,25] Data to

Table 1. The *All of Us* Guide for diversity and inclusion.

| Diversity Category | Represented in Biomedical Research | Underrepresented in Biomedical Research |
|---|---|---|
| Race and Ethnicity[a] | Individuals who identify as White and non-Hispanic | Individuals who identify as other than White and non-Hispanic (i.e. Asian; Black, African or African American; Hispanic, Spanish, or Latino; Native Hawaiian or Pacific Islander; Middle Eastern or North African) |
| Access to Care | Individuals who have had a needed medical visit in the past 12 months or can obtain and pay for medical care as needed | Individuals who have not had a needed medical visit in the past 12 months or cannot easily obtain or pay for medical care as needed |
| Age[b] | Adults ages 18–64 | Children 17 or younger and adults 65 or older |
| Annual Household Income | Individuals with household incomes above 200% of the Federal Poverty Level | Individuals with household incomes equal to or below 200% of the Federal Poverty Level |
| Disability | Individuals without a physical or cognitive disability | Individuals with either a physical or cognitive disability |
| Educational Attainment | Individuals with a high school degree or equivalent | Individuals with less than a high school degree or equivalent |
| Gender Identity[a] | Individuals who identify as either a man or a woman | Individuals who identify as gender variant, non-binary, transgender, or something else |
| Geography | Individuals who reside in urban metropolitan areas | Individuals who reside in rural and non-metropolitan areas |
| Sex Assigned at Birth[a] | Male or female individuals | Individuals who are neither male nor female (i.e. intersex) |
| Sexual Orientation | Individuals who identify as straight | Individuals who identify as asexual, bisexual, gay or lesbian or something else |

[a] Racial and ethnic minorities and women: The NIH Revitalization Act of 1993 set forth policy that mandates the inclusion of women and minorities in NIH-funded clinical research unless their exclusion is justifiable and approved by the relevant NIH institute. The resulting policy guided the enrollment criteria developed by *All of Us*. For the purpose of this work, however, women are not considered UBR. Representation of women in NIH-funded research has steadily risen to over 50%. [4]

[b] Adults aged 65 or older: NIH Policy and Guidelines on the Inclusion of Individuals Across the Lifespan as Participants in Research Involving Human Subjects defines older adults as individuals 65 years of age or older and underrepresentation of this entire group has been confirmed by the literature. Adults 75 or older, however, may be especially underrepresented; there is additional federal guidance on ensuring their inclusion. [14]

specifically assess rurality were not readily available. Only two (13%) studies reported overrepresentation of racial and ethnic minorities compared to White participants. [23,24]

Lastly, most did not draw an intentional distinction between sex assigned at birth and gender. Data to permit evaluation of gender minority representation, however, were not collected from participants or not publicly reported for any of the fifteen studies. This was the case for sexual orientation data as well. See Fig 2 for aggregate findings from the cohort assessment and Table 1 for *All of Us* Guide for Diversity and Inclusion.

## Discussion

Most of the articles sought to illustrate major health care and research barriers impacting racial and ethnic populations, along with other underrepresented groups. The literature demonstrated that racial/ethnic minorities and other groups are typically impacted by at least one of several barriers including financial (e.g., insurance coverage, income), cognitive (e.g., health literacy, technical proficiency), language (e.g., lack of translation resources), communication and cultural factors (e.g., cultural humility, lack of trust), structural (e.g., transportation, geographic accessibility), racial and ethnic discrimination, or study design (e.g., eligibility criteria limiting participation or inclusion, failure to reach diverse groups due to recruitment limitations). [7,26–34] These barriers have significant effects on multiple groups. Access to research opportunities is critically impacted by structural barriers like transportation, but also by language, communication, and unnecessary restrictions in study design. For example, willingness to participate in research does not significantly differ between individuals who reside in rural

areas and those who live in urban locations. Rather, the limiting factor is accessibility impacted by either structural barriers or recruitment limitations. [31,34,35] Older adults are also impacted by these barriers. The literature demonstrated that older adults have been marginalized in research due to a number of reasons including not meeting eligibility criteria (e.g., clinical trial restrictions on comorbid conditions due to risk of side effects), perceived burden on study staff (e.g., longer screening process or more "high touch" retention tactics needed), and cognitive or physical impairments that affect the ability to consent or fully participate. [36–38] Similar barriers are faced by racial or ethnic minorities. [39–41] Cultural barriers, specifically lack of trust, and not being trustworthy, are defining barriers across racial or ethnic minorities, particularly Black and African American individuals, [34,42] and also across many other groups including rural populations and sexual or gender minorities. Specifically in cancer-related clinical trials, studies have shown no racial or ethnic differences in participation rates [43], but this does not seem to be generally applicable to other types of clinical research settings. [44] Importantly, overcoming a lack of trust is through engagement and carefully designed and strategies [45] by which participants are invited to participate, in a respectful way, in research programs.

Representation of underrepresented groups is significantly affected by the lack of consistent demographic data collection and gaps in research domains specific to the health disparities that impact them. For example, unlike sex assigned at birth, which is a biological construct, gender is a construct that includes many dimensions. Domain experts recommend researchers include measures to capture both an individual's assigned sex at birth and gender identity to fully represent one's physical and internal sense of self. [46] The literature confirmed that gender identity as a social determinant of health is not only understudied but also unrecognized. [47,48] This body of literature is comprised of articles on evaluation of instruments that gather this information (e.g., questionnaires, electronic medical records, etc.), their limited use within healthcare and research [49,50] and the lack of knowledge and sensitivity of medical personnel about different gender identities. [51] Similarly, sexual orientation measures are not often collected from patients and research participants, especially noted for older adults. [52,53] These omissions were observed in our own review of national cohorts. The literature also noted that biomedical research has generally neglected exploration of sexual orientation, apart from research focused on understanding sexually transmitted disease. This implies a substantial gap in research that could lead to improvements in broader health outcomes for sexual and gender minorities. [54,55] The lack of data collection measures and demographic reporting does not allow us to confirm whether individuals that identify as sexual or gender minorities are truly underrepresented. Data collection is a mandatory first step, however, in addressing the obvious gaps in research results that directly impact health disparities faced by this population. For this reason, sexual and gender minorities are included as an underrepresented group.

We identified 12 articles that examined sex as a biological factor in biomedical research. Several of the articles call for more comparison of research findings by sex, noting that, like race and ethnicity, sex differences in many domains are still widely understudied. [56–60] Regardless of actual representation of females in biomedical research, many health outcomes specific to them will remain unknown unless researchers prioritize exploring differences across sex assigned at birth, race, and ethnicity. In addition, sex is predominantly perceived as a binary biological construct and individuals born as intersex, or those with disorders of sexual development, have difficulties "fitting" into a male/female binary category. Including intersex as a biological construct is important for health care since individuals' needs may not apply to either the male or female classification. [59] The literature indicates that this is also true for transgender individuals who have undergone gender affirming surgeries or who receive hormonal treatments as well. [61]

We identified literature that explained underrepresentation of select minority populations due to issues in how race and ethnicity data are traditionally collected and reported. [62,63] For example, there has been confusion among survey respondents over the categorization of Hispanic or Latino as an ethnicity rather than a race. [63] A lack of sufficient categories for race and ethnicity data, often in combination with underrepresentation within the research samples, means results are often reported in aggregated categories that can mask underlying disparities. For example, Middle Eastern and North African populations have often been classified as White in research questionnaires that do not specifically call out a category for Middle Eastern and North African, similarly Asian American, Native Hawaiian, and other Pacific Islanders have historically been grouped as Asian or Other. [62,63]

## Limitations of this approach

The broad scope and depth of this review was limited since complex strategies were needed to canvass the literature, returning exceptionally large retrieval sizes. This work is meant as a survey of trends, relying mainly on reviews obtained from the literature to best inform the state of the science on groups historically underrepresented in biomedical research. Our evaluation of other large national cohort studies was also limited because we conducted most of this work using publicly available sources. In some instances, we were unable to confirm the accuracy or currency of the information. In addition, *All of Us* is a hypothesis and disease-agnostic research program with minimal inclusion/exclusion criteria. Most of the studies we identified in our evaluation differ from *All of Us* in both areas. For example, clinical trials-enrolled participants with specific medical conditions or those that may benefit from a therapeutic treatment. They may, by their very nature, exclude certain demographic groups. Finally, many studies do not collect comprehensive socio-demographic data, therefore, we do not know whether their populations include groups we assert to be underrepresented.

## Commentary

There is a rich diversity within the U.S. that must be leveraged for understanding factors contributing to health and disease for the entire population. Historically, biomedical research has neglected this diversity to our detriment. A recent report [64] showed that, of the drugs removed from the U.S. market between 1997 and 2000, 80% were removed due to side effects or fatalities that occurred in women. Another report [65] found that racial factors, although important, do not solely account for the disparities in health care. Societal and environmental factors are equally important to account for context. [66] A widely-used antiplatelet drug, clopidogrel, needs to be converted to an active form to be effective. [67] However, due to a genetic variation that affects the body's ability to metabolize it, only about 50% of individuals of Asian ancestry are able to benefit, along with other individuals who carry the variant. This is one example of a modern scientific advancement, in this case a life-saving medication, offering a high benefit for some individuals within populations, but significant risks for others. [68]

An early research opportunity for *All of Us* will be examining health disparities among these racial and ethnic populations intersected across other demographic groups and their impact on social determinants of health. For a summary of significant conclusions, see Table 2. The *All of Us* Research Program Guide for Diversity and Inclusion (Table 1) is an essential product for *All of Us*, serving as the program foundation for 1. future demographic enrollment monitoring, 2. gauging the success of new recruitment strategies as they are implemented and providing opportunity to revise when key enrollment gaps are identified, and 3. gauging the variability and strength of the resulting research data repository, and ensuring that it will support health disparity research for all historically underrepresented groups as it grows

**Table 2. Key points learned in developing the *All of Us* guide for diversity and inclusion.**

• The rich, cultural, geographic, and demographic diversity of the U.S. is underrepresented in U.S. biomedical research.

• Significant barriers impact representation of key populations. The biomedical research community must acknowledge and address them to ensure inclusivity.

• A lack of data collection for many key demographic characteristics and social determinants is a critical gap across the entirety of biomedical research. For example, educational attainment and many other socio-economic status factors have a robust relationship with overall mortality. Yet, these measures are almost completely absent from the research landscape. *All of Us*, and the biomedical research community at large, must address this.

• Sexual and gender minority measures, except for research exploring sexually transmitted disease, have mostly been ignored. Although a lack of measures does not necessarily suggested underrepresentation, disparities impacting these groups are understudied. To understand representation of these groups, *All of Us* and the biomedical research community must start asking about sexual orientation and gender identity.

• A guide for inclusion is necessary to ensure equity in research outcomes. The guide, however, must be viewed as a starting point, not a final product in understanding and addressing barriers that impact participation, can we position *All of Us* to better understand and tackle health disparities for all.

over time. This guide is a novel attempt to emphasize and define diversity and inclusion for *All of Us* based on a series of biologically and socially significant constructs. There are considerable nuances that impact diversity and inclusion monitoring and we respect that they should not be treated independently of one another or considered universally equal. For example, there is an enormous body of literature confirming the importance of racial and ethnic diversity in biomedical research. The *All of Us* guide emphasizes minority representation in biomedical research, and the additional demographic characteristics and social determinants highlighted are meant to complement, not dilute, that inclusion.

## Conclusion

Research that accounts for wider demographic variability is critical. It is time that we move beyond relying on research generated from largely homogeneous populations of usually White, male, urban, and higher socio-economic status. [69] Only by incorporating more diverse factors as key variables, ensuring inclusion, and identifying and addressing barriers that limit research participation, can we position *All of Us* to better understand and adequately tackle health disparities.

Research is used to provide the basis by which we discover new pathways and identify new treatment approaches. As such, only by narrowing the gaps of health disparities and increasing our understanding of how social determinants of health impact health outcomes can we realize a future of research that is more precise. We note that discrimination is a standing issue in workforce diversity, [70] which impacts minority participation in research studies like ours. Based on the work presented, and in conjunction with community partners, *All of Us* has developed resources for engaging diverse communities (https://www.joinallofus.org/en/community/community-resources). These provide a springboard for researchers to more readily incorporate diversity and inclusion strategies to local settings [45,71] or disease specific contexts, or to use the https://www.researchallofus.org/ platform to build diverse cohorts. Within *All of Us*, the attention paid to these important factors can help surface the intersectionality of these barriers to research participation and help the research community to be better equipped to engage diverse populations.

The approach taken by *All of Us* was broad, methodical, and intentional in conceptualizing an inclusive guide. Despite the limitations, it encompasses a depth of inclusion that is unprecedented in other national cohorts. The emphasis on historically underrepresented minorities and other populations provides a reference for any future study that aims to ensure diversity

and equity. The *All of Us* Research Program is well on its way to creating a national resource that reflects and supports the broad diversity of the U.S. necessary for advancing precision medicine for all.

## Supporting information

**S1 Table. Search strategies by diversity category and January 2020 retrieval counts.**
(DOCX)

**S2 Table. Number of articles screened and selected in original literature search.**
(DOCX)

**S1 File. Bibliography of selected references by search category.**
(DOCX)

**S2 File. Bibliography of selected references by search category.**
(DOCX)

## Acknowledgments

We thank our colleagues, David R. Williams, Scott Sutherland, James McClain, Holly Garriock, Ed Ramos, Aditi Reddy, Stephanie Devaney, and Kelly Gebo for their support and input on the Diversity and Inclusion guide. We thank Rebecca N. Jerome for providing input on the methodology used for the project and Carolyn Diehl for assistance collecting data for the cohort assessment. We also thank the PMI NIH Protocol Development group, the DRC Pilot Research Core, and the *All of Us* PPI Committee for their efforts developing, evaluating, and finalizing the demographic measures included in the *All of Us* Basics survey. The *All of Us* Research Program would not be possible without the partnership of contributions made by its participants. See the supplementary information (S1 File) for a roster of past and present *All of Us* principle investigators. To learn more about the *All of Us* Research Program's research data repository, please visit https://www.researchallofus.org/.

## Author Contributions

**Conceptualization:** Brandy M. Mapes, Christopher S. Foster, Mona AuYoung, Karl Surkan, Robert M. Cronin, Consuelo H. Wilkins, Eliseo J. Pérez-Stable, Joshua C. Denny, Joni L. Rutter.

**Formal analysis:** Christopher S. Foster, Sheila V. Kusnoor, Marcia I. Epelbaum, Mona AuYoung.

**Methodology:** Brandy M. Mapes, Christopher S. Foster, Sheila V. Kusnoor, Marcia I. Epelbaum, Mona AuYoung, Robert M. Cronin, Joshua C. Denny, Joni L. Rutter.

**Project administration:** Christopher S. Foster, Gwynne Jenkins, Maria Lopez-Class, Dara Richardson-Heron, Ahmed Elmi, Eric Dishman, Joni L. Rutter.

**Visualization:** Brandy M. Mapes.

**Writing – original draft:** Brandy M. Mapes, Sheila V. Kusnoor, Marcia I. Epelbaum, Mona AuYoung, Maria Lopez-Class, Joshua C. Denny, Joni L. Rutter.

**Writing – review & editing:** Brandy M. Mapes, Christopher S. Foster, Sheila V. Kusnoor, Marcia I. Epelbaum, Mona AuYoung, Gwynne Jenkins, Karl Surkan, Robert M. Cronin, Consuelo H. Wilkins, Eliseo J. Pérez-Stable, Joshua C. Denny, Joni L. Rutter.

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
