## [Decision Letter · Decision Letter 0]

3 Apr 2020

PONE-D-19-34035

Diversity and Inclusion for the All of Us Research Program: A Scoping Review

PLOS ONE

Dear Mrs. Mapes,

Thank you for submitting your manuscript to PLOS ONE. After careful consideration, we feel that it has merit but does not fully meet PLOS ONE’s publication criteria as it currently stands. Therefore, we invite you to submit a revised version of the manuscript that addresses the points raised during the review process.

As you can see the reviewer comments are minor, given that your paper was well-written and informative.

We would appreciate receiving your revised manuscript by May 15 2020 11:59PM. To enhance the reproducibility of your results, we recommend that if applicable you deposit your laboratory protocols in protocols.io, where a protocol can be assigned its own identifier (DOI) such that it can be cited independently in the future. For instructions see: http://journals.plos.org/plosone/s/submission-guidelines#loc-laboratory-protocols

We look forward to receiving your revised manuscript.

Kind regards,

Dr Emma Louise Giles

Academic Editor

PLOS ONE

Journal Requirements:

2. One of the noted authors is a group or consortium"the All of Us Research Program". In addition to naming the author group, please list the individual authors and affiliations within this group in the acknowledgments section of your manuscript. Please also indicate clearly a lead author for this group along with a contact email address.

Reviewers' comments:

Reviewer's Responses to Questions

**Comments to the Author**

1. Is the manuscript technically sound, and do the data support the conclusions?

Reviewer #1: Yes

Reviewer #2: Yes

2. Has the statistical analysis been performed appropriately and rigorously? 

Reviewer #1: N/A

Reviewer #2: Yes

3. Have the authors made all data underlying the findings in their manuscript fully available?

Reviewer #1: Yes

Reviewer #2: Yes

4. Is the manuscript presented in an intelligible fashion and written in standard English?

Reviewer #1: Yes

Reviewer #2: Yes

5. Review Comments to the Author

Reviewer #1: Manuscript number: PONE-D-19-34035

Title: Diversity and Inclusion for the All of Us Research Program: A Scoping Review

Overall evaluation: In this well-written, well-organized, and concise paper, the authors present the results of a scoping review investigating the demographic characteristics of participants enrolled in US biomedical research cohorts in order to identify what groups are underrepresented in biomedical research. The results are unsurprising but present a snapshot of the lack of diversity across multiple sociodemographic characteristics within biomedical research. The authors present a set of guidelines for specific groups that researchers should make exceptional efforts to represent in biomedical research.

This is an important contribution to the literature in that it provides a moment in time in which researchers can see how we have failed to serve many underrepresented groups in research, and a call for rectifying this as we build future cohorts. While I understand that the focus of these guidelines are on population-based research, I appreciate that the authors also include some information about the lack of representation of women and racial and ethnic minority groups in clinical trials research, which has a tremendous impact on the safety of drugs and procedures for diverse groups in the US.

I have only one strong recommendation for the authors. There is a brief mention of the role of lack of trust in the low participation rate of Black and African American participants in biomedical research. This is concerning, as there is a growing body of evidence (e.g., https://doi.org/10.1002/cncr.28483) that when there is equal access to participate in and individuals are explicitly ASKED to participate in clinical trials, African Americans are no less likely to agree to participate nor to be willing to participate in clinical trials research. I fear that not including this in the current manuscript may irresponsibly perpetuate the lack-of-trust excuse that researchers often use to justify a lack of diversity in their research. We often characterize groups as “hard-to-reach” for any number of reasons, when in reality what we are saying is that they are just “easy to ignore.” I believe that the authors of this manuscript are working to change this norm in biomedical research, so I think that it is important to recognize how citing trust plays a role. Similarly, I think it is important to include any evidence we have for whether when other underrepresented groups – those from rural areas, those with disabilities, etc. -- are offered equal access to clinical trials they will accept, but this may be an important point to include in this paper to support the authors’ efforts to shift norms towards including UBR groups in research.

Other major comments:

1. Can you provide an explanation for how “Individuals who have had a needed medical visit in the past 12 months” is representative of access to care? Does this mean that they went to a medical facility, therefore they have access to care? It was unclear to me how this represents a lack of access to care.

2. In Figure 1, it is very difficult to distinguish the lines in the charts (and therefore the reported proportions of race and ethnicity) because of the colors. Also, why does the projected population of Native Hawaiian and Pacific Islanders reach 0%? Are they eliminating that category from the census?

3. Are there any actual study participants included in the Participant Provided Information Committee, particularly those representing any of the groups in the proposed guidelines? If so, this could certainly be a strength to include in the manuscript.

Minor comments:

1. All of the figures and tables are difficult to read, even as the TIFF files.

2. Line 188 need space between “and” and “170”

3. Line 209, 220, 224, possibly other instances: Data are plural, so data “were” not “was”

Reviewer #2: This article is very well written.

6. PLOS authors have the option to publish the peer review history of their article (what does this mean?). If published, this will include your full peer review and any attached files.

Reviewer #1: Yes: Katherine S. Eddens

Reviewer #2: No

---

## [Author Response · Author response to Decision Letter 0]

15 May 2020

Title: Diversity and Inclusion for the All of Us Research Program: A Scoping Review

Manuscript number: PONE-D-19-34035

Reviewer #1: I have only one strong recommendation for the authors. There is a brief mention of the role of lack of trust in the low participation rate of Black and African American participants in biomedical research. This is concerning, as there is a growing body of evidence (e.g., https://doi.org/10.1002/cncr.28483) that when there is equal access to participate in and individuals are explicitly ASKED to participate in clinical trials, African Americans are no less likely to agree to participate nor to be willing to participate in clinical trials research. I fear that not including this in the current manuscript may irresponsibly perpetuate the lack-of-trust excuse that researchers often use to justify a lack of diversity in their research. We often characterize groups as “hard-to-reach” for any number of reasons, when in reality what we are saying is that they are just “easy to ignore.” I believe that the authors of this manuscript are working to change this norm in biomedical research, so I think that it is important to recognize how citing trust plays a role. Similarly, I think it is important to include any evidence we have for whether when other underrepresented groups – those from rural areas, those with disabilities, etc. -- are offered equal access to clinical trials they will accept, but this may be an important point to include in this paper to support the authors’ efforts to shift norms towards including UBR groups in research.

Response: Thank you for raising this point as it is important to highlight. We acknowledge as well that there is evidence supporting the argument that black and African American individuals are not less likely to participate in clinical trial research, particularly cancer trials, when explicitly asked. Clinical trials, however, are associated with better care and more potential for therapeutic benefit than other types of research activities. We believe there are two issues with trust to note – there is a lack of trust (participant) and not being trustworthy (researchers/institutions). Inviting individuals to participate, in a respectful way, may aid in overcoming the lack of trust barrier. This may not mean, however, that the barrier does not exist. We have addressed this point and clarified in the manuscript that the reason to cite low trust is so that researchers become more trustworthy (https://www.ncbi.nlm.nih.gov/pmc/articles/PMC6143205/) and that racial differences in trust are not just historically important to note (https://www.ncbi.nlm.nih.gov/pmc/articles/PMC6802409/). 

Reviewer #1: Can you provide an explanation for how “Individuals who have had a needed medical visit in the past 12 months” is representative of access to care? Does this mean that they went to a medical facility, therefore they have access to care? It was unclear to me how this represents a lack of access to care.

Response: Access to care can be measured according to several major elements including timeliness of receiving care when a need arises. This element is often measured according to one’s ability to receive care when needed over the course of a 12 month timespan.* This means the individual had the resources required to receive care when needed (ex. Transportation, money, locality to a medical care clinic, etc.) The All of Us Research Program recommends that both individuals who have not received a needed medical visit in the past 12 months or cannot easily obtain or pay for care as needed should be considered underrepresented in biomedical research. 

*1. Elements of Access to Health Care | Agency for Health Research and Quality [Internet]. [cited 2020 Apr 19]. Available from: https://www.ahrq.gov/research/findings/nhqrdr/chartbooks/access/elements.html

Reviewer #1: In Figure 1, it is very difficult to distinguish the lines in the charts (and therefore the reported proportions of race and ethnicity) because of the colors. Also, why does the projected population of Native Hawaiian and Pacific Islanders reach 0%? Are they eliminating that category from the census? 

Response: Figure 1 has been revised so that the lines are wider and easier to distinguish. Please note that the project population of Native Hawaiian and Pacific Islanders are as follows: 0.2% (2016), 0.2% (2020), 0.3% (2030), 0.3% (2040). The figure originally submitted listed the 2020 population as 4% in error. 

Reviewer #1: Are there any actual study participants included in the Participant Provided Information Committee, particularly those representing any of the groups in the proposed guidelines? If so, this could certainly be a strength to include in the manuscript.

Response: Yes, more than 30 participant representatives are active members of the All of Us governance committees. The Participant Provided Information Committee roster includes two participant partners. This is now noted in the manuscript. 

Reviewer #1: Minor comments:

1. All the figures and tables are difficult to read, even as the TIFF files.

2. Line 188 need space between “and” and “170”

3. Line 209, 220, 224, possibly other instances: Data are plural, so data “were” not “was”

Response: Thank you for this feedback.

1. The figures and tables have been revised to improve readability. Figure 3 has been repurposed as a table (Table 3). 

2. This grammatical error has been corrected.

3. These corrections have also been made.

---

## [Editor Report · Decision Letter 1]

8 Jun 2020

Diversity and inclusion for the All of Us Research Program: a scoping review

PONE-D-19-34035R1

Dear Dr. Mapes,

We are pleased to inform you that your manuscript has been judged scientifically suitable for publication and will be formally accepted for publication once it complies with all outstanding technical requirements.

With kind regards,

Emma Louise Giles

Academic Editor

PLOS ONE

Additional Editor Comments (optional):

Thank you for revising your manuscript and considering all of the reviewer comments. The final minor comment I wish to make is that there are formatting errors on Page 15, which may need to be corrected. Thank you for such an interesting manuscript.
---

## [Editor Report · Acceptance letter]

18 Jun 2020

PONE-D-19-34035R1 

Diversity and inclusion for the *All of Us *Research Program: a scoping review 

Dear Dr. Mapes:

I'm pleased to inform you that your manuscript has been deemed suitable for publication in PLOS ONE. Congratulations! Your manuscript is now with our production department. 

Kind regards, 

on behalf of

Dr. Emma Louise Giles 

Academic Editor

PLOS ONE